# Type I Interferons in Systemic Lupus Erythematosus: A Journey from Bench to Bedside

**DOI:** 10.3390/ijms23052505

**Published:** 2022-02-24

**Authors:** Tao Ming Sim, Siying Jane Ong, Anselm Mak, Sen Hee Tay

**Affiliations:** 1Department of Medicine, Yong Loo Lin School of Medicine, National University of Singapore, Singapore 117597, Singapore; e0268883@u.nus.edu (T.M.S.); mdcam@nus.edu.sg (A.M.); 2Division of Rheumatology, Department of Medicine, National University Hospital, Singapore 119074, Singapore; jane.ong@mohh.com.sg

**Keywords:** systemic lupus erythematosus, SLE, interferon, IFN, biologics, anifrolumab, precision medicine

## Abstract

Dysregulation of type I interferons (IFNs) has been implicated in the pathogenesis of systemic lupus erythematosus (SLE) since the late 1970s. The majority of SLE patients demonstrate evidence of type I IFN pathway activation; however, studies attempting to address the relationship between type I IFN signature and SLE disease activity have yielded conflicting results. In addition to type I IFNs, type II and III IFNs may overlap and also contribute to the IFN signature. Different genetic backgrounds lead to overproduction of type I IFNs in SLE and contribute to the breakdown of peripheral tolerance by activation of antigen-presenting myeloid dendritic cells, thus triggering the expansion and differentiation of autoreactive lymphocytes. The consequence of the continuous stimulation of the immune system is manifested in different organ systems typical of SLE (e.g., mucocutaneous and cardiovascular involvement). After the discovery of the type I IFN signature, a number of different strategies have been developed to downregulate the IFN system in SLE patients, finally leading to the successful trial of anifrolumab, the second biologic to be approved for the treatment of SLE in 10 years. In this review, we will discuss the bench to bedside translation of the type I IFN pathway and put forward some issues that remain unresolved when selecting SLE patients for treatment with biologics targeting type I IFNs.

## 1. Introduction

Systemic lupus erythematosus (SLE) is an autoimmune disease characterized by complex, heterogeneous clinical manifestations, involving the skin, vessels, kidneys and central nervous system [1]. The disease course is also unpredictable, with remissions and flares that lead to cumulative organ damage and mortality [2]. The female to male incidence of SLE varies with age, being approximately 1 during the first decade of life and peaks at 9 during the 4th decade, afflicting women of childbearing age [3]. The prevalence of SLE has been increasing over time, from 40 per 100,000 in the 1970s to 100 per 100,000 since the 2000, while the incidence has been relatively constant, ranging from 4.8 to 7.2 per 100,000 [4]. This numerical discrepancy between prevalence (cross-sectional estimate of the number of cases per 100,000 of the population per year) and incidence (number of new cases per 100,000 of the population per year) likely reflects the improved diagnosis and survival rates over time, as well as the lifelong nature of this disease [5,6]. While the survival rates of SLE patients have increased over the past 5 decades, organ damage, particularly to the renal and neuropsychiatric systems, has hindered further improvement in survival [2]. Organ damage may also be attributed to the chronic use of glucocorticoids, leading to increased cardiovascular events, osteonecrosis and osteoporosis with fractures [5]. This is undesirable, as glucocorticoids remain the mainstay of SLE therapy since its approval by the US Food and Drug Administration (FDA) in 1955 [6].

There have been six medications approved by the US FDA for the treatment of SLE: aspirin, hydroxycholoroquine, glucocorticoids, belimumab, voclosporine and, most recently, anifrolumab in 2021 [6,7,8,9]. The approval of anifrolumab, the second biologic to be approved for SLE in a decade, marks a major advance in the treatment of SLE [10]. Belimumab, an anti-B lymphocyte stimulator monoclonal antibody, was the first biologic to be approved for the treatment of active, autoantibody-positive SLE in 2011 [11,12]. Belimumab’s efficacy in the treatment of active lupus nephritis (LN) led to further FDA approval of this indication [13]. Anifrolumab is a fully human, IgG1κ monoclonal antibody that binds to type I interferon (IFN) receptor subunit 1 (IFNAR1) to suppress signaling by all type I IFNs [14,15]. This is particularly befitting with the understanding of SLE being a prototype “type I interferonopathy” [16]. However, the type I IFN landscape has also been littered with a number of negative clinical trials targeting type I IFNs with various agents, including that of anifrolumab itself [17,18,19]. SLE remains a drug development challenge due to its clinical heterogeneity complicating clinical trial design and end point selection [1,20]. In this review, we will discuss the bench to bedside translation of the type I IFN pathway and put forward some issues that remain unresolved when selecting SLE patients for treatment with biologics targeting type I IFNs.

## 2. Interferon Pathways Leading to SLE

The first step in determining that SLE is a disease of the immune system was taken in 1948 when Hargraves et al., discovered the lupus erythematosus cell phenomenon [21]. In 1957, Issacs and Lindenmann discovered that in cell culture, heat-inactivated influenza virus induces a soluble factor that inhibits propagation of live influenza virus; they named this factor IFN [22]. SLE is characterized by dysregulation in both the innate and adaptive immune systems, and type I IFNs serve as a class of cytokines that bridges innate and adaptive immunity [23]. Traditionally, SLE has been considered as a disease of perturbed adaptive immunity due to the critical pathogenic roles of T and B cells [24,25]. Chronic type I IFN production in SLE shifts the differentiation of naïve CD4^+^ T cells from a Th1 effector subset toward a dominant T follicular helper cell phenotype, promoting B cell differentiation, immunoglobulin class switching, affinity maturation and ultimately leading to the secretion of antinuclear antibodies (ANAs), which are an immunological hallmark of SLE [26,27]. The two main types of ANAs, anti-DNA and anti-RNA-binding protein (RBP) antibodies are secreted by plasmablasts and plasma cells, respectively [27]. Interestingly, autoantibodies specific to RBPs (Ro, La, Sm or U1 RNP), but not anti-DNA, were independently associated with high expression of INFα-inducible genes in the peripheral blood mononuclear cells (PBMCs) of SLE patients [28]. Of the anti-RBPs, anti-U1 RNP is the strongest predictor of IFN gene signature of expression in whole blood samples of SLE patients of both African and European ancestries [29].

### 2.1. History of Type I IFNs in SLE

It was not until 1969 that the notion that type I IFNs might play a role in the immunopathogenesis of SLE was raised by Steinberg et al. [30]. Steinberg et al., described acceleration of disease in the NZB/NZW murine lupus model following the administration of polyinosinic-polycytidylic acid (poly IC), an inducer of type I IFNs [30]. Hooks et al., first reported high titers of IFN in the serum of SLE patients in 1979, and this finding was later confirmed to be mainly due to IFNα by Preble et al., in 1982 [31,32]. In 2003, several independent laboratories simultaneously reported on the use of microarray analysis of gene expression in the peripheral blood of pediatric and adult SLE patients to demonstrate a striking overexpression of gene transcripts in the IFN pathway, termed the “type I IFN signature” [33,34,35,36]. A recent meta-analysis of 16 datasets comprising 190 samples derived from primary human cells treated with type I IFN was performed to obtain a robust set of type I IFN-stimulated genes [37]. The same paper also described a unique 93-gene signature (SLE MetaSignature) from 40 independent studies that distinguishes SLE from other autoimmune, inflammatory and infectious diseases and that persists across diverse tissues and cell types [37]. Of the 93 genes, 70 were differentially expressed in primary cells stimulated by type I IFN [37]. In keeping with this signature, IFNα therapy in cancer and viral infections induces autoantibody formation in 4–19% of patients and a variety of SLE-like symptoms have been reported in 0.15–0.7% of them [38]. In addition, monogenic interferonopathies such as Aicardi Goutières syndrome share some similarities with the polygenic forms of SLE [39].

Interferons are assigned to one of three families: type I, type II or type III [40]. We now know that multiple species of type I IFNs exist; these can be divided into five classes (IFN-α, -β, -ε, -κ and -ω), of which IFNα can be further subdivided into 13 classes (IFN-α1, -α2, -α4, -α5, -α6, -α7, -α8, -α10, -α13, -α14, -α16, -α17 and -α21) [41,42]. The type II IFN family consists of one IFNγ and the type III IFN family comprises of IFNλ1, IFNλ2, IFNλ3 and IFNλ4 [40]. The terms “type I IFN signature” or “IFNα signature” are used in the literature to distinguish the IFN signature mentioned above from those induced by type II and III IFNs [23]. Type I IFNs all bind to the same ubiquitously expressed type I IFN receptor (IFNAR) that consists of two polypeptide chains, IFNAR1 and IFNAR2, with IFNβ having a higher affinity for IFNAR than IFNα [26,43]. Canonical IFNAR signaling depends on the Janus kinase 1, tyrosine kinase 2, signal transducer and activator of transcription (STAT) 1, STAT 2 and IFN regulatory factor 9 to induce new gene transcription to mediate antiviral responses [40]. The levels of type I IFNs peak in the first few days after acute viral infections, a response that is time limited, normalizing when the virus is cleared [40]. However, a notable feature in SLE is that the type I IFN pathway is activated over time, which may indicate a significant heritable contribution to the disease [26,44]. For example, it has been demonstrated that high serum IFNα activity is frequently found in healthy family members of SLE patients compared to healthy unrelated donors and high INFα activity is clustered in certain families among SLE patients and their first degree relatives [44]. In addition, autoantibodies to DNA and RBP were very uncommon in healthy family members, hence the IFN pathway activation was not caused by immune complex stimulation in this setting [44]. IFN-related genetic variants such as *IRF5*, *IRF7*, *IRF8*, *STAT4*, *PTPN22*, *OPN*, *IFIH1* and *TYK2* playing an important role in SLE pathogenesis have been identified [45]. In summary, these lines of evidence suggest that genetic variations in addition to the type I IFN pathway are required to lower the threshold for immune activation and development of autoantibodies in individual SLE patients [26].

### 2.2. Contribution of Type II and III IFNs to SLE Immunopathogenesis

Advancement in technology has allowed more in-depth gene expression studies to shed light on the molecular pathogenesis of SLE, starting from microarray platforms to RNA sequencing and, more recently, single-cell RNA sequencing [16,33,34,35,36,46]. As the technology platforms grew in sophistication, it became important to develop novel strategies to analyze such large scale data [47]. Chaussabel et al., designed a modular-analysis framework that is based on the identification of transcriptional modules formed by genes coordinately expressed in multiple disease datasets [47]. A module is formed of transcripts belonging to the same clusters across diseases [47]. Using this approach, three IFN modules (M1.2, M3.4 and M5.12) were identified in 87% of whole blood samples from adult SLE patients [48]. Strikingly, the IFN signature was more complex than expected, with each module displaying a distinct activation threshold (M1.2 < M3.4 < M5.12) [48]. When only one of the three IFN modules was upregulated, it always corresponded to M1.2 [48]. M3.4 appeared next and there was no M5.12 upregulation in the absence of the other two [48]. Mining of other datasets identified that IFNα upregulated to M1.2, while M3.4 and M5.12 could be driven by INF-β and -γ [48]. It is now appreciated that SLE patients with active disease have elevated levels of circulating type I, II and III IFNs and that different organ involvement seems to be related to different IFN types [49,50]. There is significant overlap between the genes induced by type I, II and III IFNs, and different investigators may choose to measure different IFN-related genes via reverse transcription polymerase chain reaction (RT-PCR) [43,50]. Hence, the results have been inconsistent and sometimes challenging to interpret as there is no consensus on how to define the IFN score today [43].

### 2.3. Physiological Role of Type I IFNs in Viral Infections

Depending on the type of stimulus, type I IFN production can be induced in a broad range of cells types. IFNα production is limited to mainly myeloid cells such as plasmacytoid dendritic cells (pDCs), monocytes and, as are increasingly recognized, neutrophils [51,52,53]. One key aspect of type I IFN biology is its ability to act as an innate antiviral cytokine, which leads to the establishment of an antiviral state, characterized by expression of many proteins involved in the suppression of viral replication and spread, including proteins involved in RNA degradation, translational inhibition and cellular apoptosis [54]. One example is the dsRNA-activated protein kinase R (PKR). The transcription of *EIF2AK2* coding for PKR is upregulated by type I IFN signaling, and the binding of dsRNA produced during viral replication alters the conformation of PKR, which leads to dimerization and activation by autophosphorylation [55]. Once activated, PKR phosphorylates the α-subunit of eukaryotic initiation factor 2 to inhibit protein translation and suppress viral replication [55]. Other IFN-stimulated transcripts important for antiviral response include *MX1*, *APOBEC1* and the family of *IFITM* and *TRIM* genes [56]. The importance of type I IFNs in the role of viral infections is highlighted in the recent work by Bastard et al., whereby neutralizing autoantibodies against all 13 types of INFα, IFNω or both were demonstrated in the plasma of patients with severe COVID-19 pneumonia [42]. This phenomenon of anti-IFN autoantibodies has also been observed in SLE patients, with the presence of de novo or induced anti-IFNα autoantibodies that normalized the type I IFN signature [19,57]. Interestingly, viral infections such as human immunodeficiency virus (HIV) lead to chronic activation of the type I IFN pathway [40]. In fact, the immunopathogenic mechanisms described in HIV-infected patients are similar to those of SLE [26].

### 2.4. Interferon System and Disease Manifestations in SLE

Specific clinical manifestations are apparently related to different types of IFN. For instance, high IFNα was noted to be associated with mucocutaneous manifestations including chronic discoid lesions [58] while IFNγ was associated with high SLE Disease Activity Index (SLEDAI) score and the occurrence of LN. High IFNλ1 was noted to be related to anti-nucleosome antibodies and higher frequency of anti-phospholipid antibodies [50]. Increased IFN transcripts were noted in patients with musculoskeletal and cutaneous manifestations of SLE, elevated ESR and serum anti-dsDNA level and low serum complement level [59]. Chronic lupus erythematosus, acute and subacute cutaneous lupus and photosensitivity are associated with increased type I IFN signature. In addition, patients with subacute cutaneous lupus and discoid lupus were shown to have increased IFN signature, which correlated with increased activity of the skin [60]. However, the changes in IFN signatures were not associated with changes in SLE disease activity over time [59].

Anaemia, leucopenia and thrombocytopenia are common in patients with SLE during the course of the illness. Type I IFNs directly suppress the bone marrow production of haematopoietic cells. Administration of anifrolumab was noted to be associated with improvement of lymphopenia, highlighting the pathophysiologically important impact of type I IFNs on the bone marrow in patients with SLE [61]. As for renal disease, pDCs, one of the pivotal sources of type I IFNs, infiltrate the kidneys and renal tubular cells in patients with LN and demonstrate type I IFN signatures [62,63]. Type I IFNs potentially assist with recruiting neutrophils in the kidneys that induce LN via IL-17 [64]. The role of type II IFN, however, was not well addressed in the context of clinical LN. Blockage of IFNγ with AMG811 did not demonstrate ameliorate of LN, nor clinical as well as serological disease activity of SLE in general [65]. SLE patients with complete renal response to treatment at 12 months had significantly lower IFN signature scores compared to those who did not reach complete remission [66]. Lastly, arthritis in patients with SLE was shown to be associated with IFNγ signatures, which is in contrast to lupus skin involvement, whereby its pathological association is with type I IFN signature [50].

## 3. Biologics Targeting Type I Interferons in SLE

In view of the central role of type I IFNs in the immunopathology of SLE, targeting the IFN pathway has been proposed as a novel treatment for SLE [67]. There has been expansive research on various modalities targeting different aspects of the IFN pathway, including monoclonal antibodies against IFNα and anti-IFNα antibody-inducing vaccines [68]. For the purpose of this review, we will be focusing on biologics that target type I IFNs (Table 1).

### 3.1. Rontalizumab

Rontalizumab is a humanized IgG1 monoclonal antibody developed as a potential biologic for the treatment of SLE with the ability to bind and neutralize all known subtypes of IFNα [69]. A phase I trial in a cohort of 60 patients with stable, mildly active SLE studied the safety and pharmacodynamic properties in rontalizumab [69]. A dose-dependent reduction in expression levels of seven pre-determined IFN-regulated genes representative of the IFN signature with single and repeat doses of rontalizumab was found. Rontalizumab was also reported as being generally safe and well-tolerated. Most of the adverse effects were mild or moderate, with the most common being upper respiratory tract infections, nausea and vomiting, headaches, musculoskeletal and connective tissue signs and symptoms and urinary tract infections. Despite the role of type I IFN in modulating host immunity, the exposure-adjusted rate of infections was found to be similar between treatment groups, with no dose-related increase in infection.

A phase II trial immediately followed consisting of two sequential placebo-controlled sub-studies [18]. This trial involved 238 patients with moderate to severe SLE with active disease as defined by the British Isles Lupus Disease Activity Group (BILAG) index: with BILAG A (severe disease activity in 1 or more domains) or BILAG B (moderate disease activity in 2 or more domains) [76]. The participants had background immunosuppression suspended and were randomized to either intervention group (750 mg rontalizumab every 4 weeks) or placebo. At week 24, no significant difference in treatment response was found as determined by the primary and secondary end points: the BILAG and SLE response indices (SRI), respectively, while no adverse safety signal was reported. Further phase III clinical trials were not undertaken in view of the lack of efficacy of rontalizumab in the phase II trial, which has been proposed to be due to the molecule’s specificity toward IFNα, leaving other type I IFNs available for binding and activation of IFNAR, mediating downstream signaling [77].

### 3.2. Sifalimumab

Following the failure of rontalizumab, the search for an effective biologic in targeting the type I IFN pathway continued. Sifalimumab is a fully humanized IgG1κ monoclonal antibody with an ability to bind to and neutralize most of the 13 known IFNα subtypes [71]. The first study of sifalimumab in patients with SLE involved a phase I randomized, double blind, placebo-controlled trial of 51 patients to study the safety profile, immunogenicity and pharmacological properties of the biologic [70]. The reported adverse effects were similar between treatment and placebo groups and were generally mild. No significant increase in viral infections was noted compared to the placebo. Importantly, this study confirmed that sifalimumab neutralized overexpression of type I IFN signature in SLE patients in a dose-dependent manner.

Multicentre phase II trials on sifalimumab were conducted on a group of 431 patients with active SLE, with the primary end point of the 52-week randomized, double-blind, placebo-controlled trial being the percentage of patients achieving an SRI(4) response at end of the 52 weeks [72]. At week 52, improvements as determined by the SRI(4) scores were found in the three dosage groups of sifalimumab [200 mg (*p* = 0.057), 600 mg (*p* = 0.094) and 1200 mg (*p* = 0.031), with *p* value of ≤0.098 considered statistically significant] compared to the placebo group. Sifalimumab was also found to result in improvement in skin score and a clinically significant reduction in swollen and tender joint counts. As a whole, this trial demonstrated clinical efficacy of IFNα inhibition by sifalimumab, as evidenced by improvements in both organ specific outcomes, including mucocutaneous, musculoskeletal, renal, haematological and vascular manifestations of SLE, and global outcomes of SLE with an acceptable safety profile. Despite the authors concluding that type I IFN blockade is a promising approach for the treatment of moderate to severe SLE and that sifalimumab had reasonable clinical efficacy, the sponsors suspended development of sifalimumab in favor of anifrolumab, a novel biologic developed by the same pharmaceutical company targeting IFNAR.

### 3.3. Anifrolumab

Anifrolumab is a fully human IgG1κ monoclonal antibody with the ability to bind to IFNAR, allowing it to inhibit the formation of IFN-IFNAR complex and downstream gene transcription [78]. In contrast to rontalizumab and sifalimumab, which were designed to bind and neutralize IFNα, anifrolumab antagonizes the receptor responsible for cellular signaling induced by all types of type I IFNs, including IFN-α, -β, -ε, -κ and -ω [18,72,78].

Safety, tolerability and pharmacokinetics of anifrolumab administered subcutaneously and intravenously were studied in 30 healthy volunteers in a phase I, single centre, double-blind randomized controlled trial (RCT) [73]. Both routes of administration were found to be well-tolerated. Fewer adverse events were reported in the placebo group than in the treatment group. Of note, no serious adverse effects were reported in the anifrolumab group, with the most common adverse effects being upper respiratory tract infection and dry throat. Subsequent phase II trials were conducted to evaluate the efficacy of anifrolumab in the treatment of SLE. The MUSE trial was a Phase IIb, double blind trial in which a cohort of 305 SLE patients with moderate to severe disease were randomized to receive IV anifrolumab (300 mg or 1000 mg) or placebo every 4 weeks for a duration of 48 weeks [74]. The subjects were stratified according to disease activity as determined by the SLEDAI-2K, their high or low IFN signature based on gene expression and oral corticosteroid dose. The primary end point of this phase II trial was the percentage of patients with an SRI(4) response at week 24 and a sustained reduction in oral corticosteroids. Compared with the placebo, a higher proportion of subjects in the treatment group (34.3% of 99 subjects in 300 mg group, 28.8% of 104 subjects in 1000 mg group) met the primary end point as compared to the placebo (17.6% of 102 subjects). Approximately 75% of participants in the trial had a high IFN signature at baseline, and a larger response was demonstrated in the IFN-high subgroup. In this subgroup, greater efficacy with anifrolumab was found as compared to the placebo at both 300 mg and 1000 mg. The response rates in subjects with a low IFN signature at baseline were similar to that in the placebo group; however, given the small sample size of the IFN-low subgroup, the interpretation of efficacy in this subset analysis might have been limited. The authors proposed future larger studies to evaluate the effects of anifrolumab in patients with a low IFN signature. By week 52 of the trial, multiple primary and secondary end points were reached in the anifrolumab group, including SRI(4), BILAG-Based Composite Lupus Assessment (BICLA), modified SRI(6) and BILAG-2004 clinical responses. Furthermore, at the end of the 52 weeks, anifrolumab-treated patients were also demonstrated to have undergone greater improvements in organ-specific disease measures and outcomes as compared to the placebo group, with a greater percentage of subjects showing improvements in skin manifestations of SLE and number of swollen and tender joints. Anifrolumab was found to be well-tolerated, and the adverse events that were reported were similar across the placebo and anifrolumab groups. Of note, a dose-related increase in the occurrence of upper respiratory tract infections and reactivation of herpes zoster was observed in the anifrolumab-treated patients. The promising results paved the way for further evaluation of anifrolumab, giving rise to the Treatment of Uncontrolled Lupus via the Interferon Pathway (TULIP) trial, which consists of two phase III trials named TULIP-1 and TULIP-2.

TULIP-1 was a multi-center, randomized, double-blind, placebo-controlled parallel-group conducted in 123 sites in 18 countries, in which 457 subjects with moderate to severe, active SLE were randomized to receive either anifrolumab 150 mg intravenously (n = 93), 300 mg intravenously (n = 180) or placebo (n = 184) in addition to a stable standard of care treatment every 4 weeks for a duration of 48 weeks [17]. Prior to randomization, subjects were stratified by a SLEDAI-2K score (<10 or ≥10), type I IFN gene signature (high or low) and a daily oral corticosteroid dose (<10 or ≥10 mg/day). The primary outcome measured was the proportion of patients who achieved an SRI(4) response at week 52, and it was found that the SRI(4) response was similar between the anifrolumab 300 mg group (36%) and the placebo group (40%). Analysis of the patients with a high IFN signature in the anifrolumab 300 mg group compared to those in the placebo group did not yield any significant differences in SRI(4) responses. These equivocal results, despite the promising results from the previous MUSE trial, led to a re-evaluation and critical analysis of the study design of the TULIP-1 trial. It was found that the original medication rules of the study classified subjects, with the new use of nonsteroidal anti-inflammatory drugs (NSAIDs) as nonresponders, were inconsistent with the intention of the protocol since NSAIDs may not be considered as crucial as other immunosuppressants, such as corticosteroids, in such trials. Medication rules were adjusted and key analyses were reperformed to allow for NSAID use up to week 50 to be classified as responders. After which, the primary end point was still found to not be met in TULIP-1. However, several key secondary outcomes were associated with improvements, including sustained oral corticosteroid dose reduction, organ-specific measures of joint and skin responses and BICLA response. The incidence of adverse effects among participants in the TULIP-1 trial was similar to that from the MUSE trial; most notably, the incidence of herpes zoster was found to be higher in the anifrolumab group (5% in 150 mg anifrolumab group, 6% in 300 mg anifrolumab group) compared to the placebo (2%), which is concordant with findings from the MUSE trial.

TULIP-2 was a separate phase III, multi-center, multinational, double-blind, placebo-controlled RCT conducted to evaluate the efficacy of anifrolumab in a group of 362 subjects with SLE [8]. Findings from TULIP-1 shaped the measured outcomes of TULIP-2: the observation that anifrolumab in SLE patients yielded clinical responses according to the BICLA response but not to SRI(4) resulted in the primary end point of TULIP-2 being stipulated as a BICLA response. Furthermore, modified medication rules were applied to TULIP-2, and patients who used NSAIDs during the study period were not classed as nonresponders. The 362 participants of the TULIP-2 trial were randomized to receive either intravenous anifrolumab 300 mg (n = 180) or the placebo (n = 182) every 4 weeks for 48 weeks. Similar to TULIP-1, randomization into study groups in TULIP-2 was stratified according to SLEDAI-2K score at screening (<10 or ≥10), type I IFN signature (high or low) and baseline oral glucocorticoid dose (<10 mg per day or ≥10 mg per day). The percentage of subjects in the anifrolumab group (47.8%) who achieved a BICLA response at the end of the study and therefore met the primary outcome of the study was significantly higher than that in the placebo group (31.5%). Various key secondary end points were also achieved. In the subpopulation of subjects with high IFN gene signature, the percentage of patients who achieved a BICLA response at week 52 was 48.0% in the anifrolumab group compared to 30.7% in the placebo group, demonstrating statistical significance (*p* = 0.002). Another critical secondary end point met was that of oral corticosteroid dosage at week 52. Out of the group of patients who were receiving prednisone equivalent to 10 mg doses or more per day at baseline, a sustained reduction in daily dose to 7.5 mg or less occurred in 51.5% of patients in the anifrolumab group compared to 30.2% of patients in the placebo group. Anifrolumab was also shown to be efficacious in significantly improving skin manifestations in patients with at least moderately active skin disease at baseline. However, numbers of swollen and tender joints and annualized flare rates did not see significant increases with anifrolumab treatment. The safety profile of anifrolumab in the TULIP-2 trial was comparable to both the MUSE and TULIP-1 trials. The incidence of herpes zoster among subjects on anifrolumab was 7.2%, similar to that in the MUSE and TULIP-1 trials. The most frequent serious adverse effect was that of pneumonia, which was recorded in three subjects in the anifrolumab group of the TULIP-2 trial.

TULIP-LN was a phase II, double-blind RCT investigating the efficacy and safety of an intravenous regimen of two different doses of anifrolumab versus the placebo in a group of 145 subjects with active, biopsy-proven, Class III or IV LN [75]. As the original TULIP-1 and TULIP-2 trials excluded patients with severe, active LN, TULIP-LN was an RCT that was designed to specifically evaluate the efficacy of anifrolumab in active LN. One hundred and forty-five subjects were randomized to receive a monthly intravenous anifrolumab basic regimen of 300mg (n = 45) and an intensified regimen of 900mg for the first three doses and 300 mg thereafter (n = 51) or the placebo (n = 49). Randomization was stratified according to the 24-h urine protein:creatinine ratio (UPCR) and type I IFN gene signature status. The primary end point of change in baseline 24-h UPCR at week 52 for combined anifrolumab versus the placebo group did not reach significanc; however, it is claimed that the results were adversely affected by the suboptimal anifrolumab exposure obtained with the basic regimen dosing. This suboptimal pharmacokinetic exposure with anifrolumab was attributed to increased clearance associated with proteinuria in LN [79,80]. The anifrolumab-intensified regimen was found to be associated with clinically meaningful responses over placebo for various secondary end points. For example, a treatment difference of 27.6% compared to the placebo for alternative complete renal response (aCRR), a stringent end point requiring CRR and inactive urinary sediment was observed. Most reported adverse effects were mild or moderate in intensity, and the safety profile of anifrolumab in LN was generally consistent with the safety profile from the TULIP-1 and TULIP-2 trials. Herpes zoster occurred in 20.0%, 13.7% and 8.2%, respectively, of patients undergoing intensified regimen, basic regimen and the placebo.

There is still an ongoing trial for anifrolumab in SLE patients, namely the TULIP SLE LTE (NCT02794285), which is a phase III, multinational, double-blind RCT in moderate to severe SLE subjects who completed TULIP-1 or TULIP-2 to characterize the long-term safety and tolerability of intravenous anifrolumab versus the placebo.

## 4. Discussion

The advent of biologics in the past few decades has revolutionized the landscape for the management of autoimmune conditions such as rheumatoid arthritis, psoriasis, and inflammatory bowel disease [77]. Unfortunately, the early clinical trials for biologics in SLE patients were not met with success, with multiple reasons, such as disease heterogeneity of SLE, selection of study endpoints, beneficial effects of background therapies and the use of rescue therapies for disease flares, impeding the ability to accurately measure efficacy of the biologic being studied [81]. SLE is a chronic autoimmune condition that can involve multiple organs and has a myriad of clinical manifestations. The heterogeneity and unpredictable disease course of SLE, coupled with difficulty in assessing the clinical response to various therapeutic agents have been hurdles in the design, development and evaluation of novel agents through clinical trials [82]. This is particularly pressing with a number of biologics targeting B cells (belimumab and daratumumab), T cells (dapirolizumab), type I IFNs (anifrolumab) and small molecules (fenebrutinib, evobrutinib, baricitinib and tofacitinib) inhibiting kinases in clinical trials and potentially receiving approval in the coming years [83].

The array of clinical phenotypes in SLE reflects the complex cellular and molecular mechanisms involved in its immunopathogenesis [83]. Several pathways or immune signatures are likely to be operational in each SLE patient, but the relative contributions vary between individuals [83]. As such, there is still an unmet need in terms of biomarker development in SLE to guide treatment. Precision medicine refers to a tailored treatment approach, with careful selection of the appropriate demographics of patient based on genetic, epigenetic and disease properties and, correspondingly, the choice of appropriate therapeutic regimes. Precision medicine aims to maximize efficacy and reduce adverse effects and has been a center of interest in the development and optimization of novel treatments for a wide spectrum of diseases [84,85]. Increased understanding of SLE and its pathogenesis has spurred much interest in targeted therapy against these crucial disease pathways, such as biologics against B cells, T cells, cytokines, costimulatory pathways and the type I IFN pathway [86]. While targeting the type IFN pathway has proven to be efficacious, further studies are warranted to determine how to select appropriate patient populations for precision medicine treatment of SLE, while at the same time considering the safety profiles and efficacy of existing and future IFN targeting modalities. It might be prudent to measure the type I IFN signature and its subtypes, together with de novo anti-IFN autoantibodies before making a decision on treatment with anifrolumab. In addition, a potential concern about targeting type I IFNs in the current COVID-19 pandemic, or in future viral pandemics, is the possibility of severe viral infections. Moreover, the increasing armamentarium of medications available for the treatment of SLE poses a dilemma as to which to use and the treatment paradigms (e.g., induction, followed by sequential additive or simultaneous additive therapies). For example, a whole blood RT-PCR classifier to classify SLE patients with predominant IFN, plasmablast, neutrophil or erythropoiesis signatures may be warranted before embarking on targeted therapy against certain cytokines or immune cells [16,87].

Ten years have passed since FDA approval of the first biologic for the treatment of SLE. We posit that the next ten years will be poised for more therapies approved for SLE, built upon the discoveries that rheumatologists and immunologists have made over the past decades (Figure 1). Coupled with the availability of high throughput multi-omics platforms and their integration using bioinformatics, we will likely witness tremendous insights into the disease pathogenesis and treatment for SLE patients [88].

## Figures and Tables

**Figure 1 ijms-23-02505-f001:**
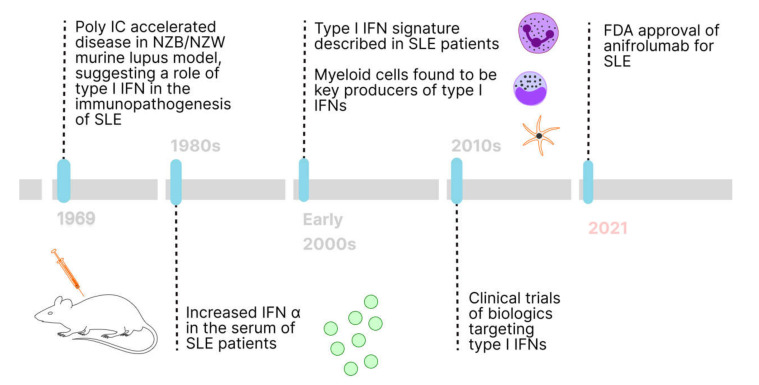
Timeline depicting the initial discovery of the role of type I IFN in SLE immunopathogenesis to present day FDA approval of biologic targeting the type I IFN pathway (anifrolumab).

**Table 1 ijms-23-02505-t001:** Summary of clinical trials of biologics targeting type I IFN in SLE.

Study, n, SLE Population	Phase	Results	Effects on IFN Signature	Significant Adverse Events
McBride et al., 2012 [69], N = 60 (Placebo n = 12 vs. Rontalizumab n = 48). Subjects with stable, mildly active SLE as defined by SELENA-SLEDAI ^a^ score. ~50% of patients had baseline high interferon signature.	I	The pharmacokinetic properties of rontalizumab were as expected for an IgG1 monoclonal antibody and were found to be proportional to dose.	At baseline, patients were categorized by type I IFN signature high or low from expression of seven selected type I IFN inducible genes. IFN-regulated genes expression demonstrated a dose-dependent decline, which was evident in the majority of patients, regardless of high or low baseline IFN signature was sustained beyond 28 days after dosing.	An acceptable safety profile was demonstrated. The incidence of serious adverse events was comparable across cohorts. These serious adverse events were classified as unrelated to rontalizumab.
Kalunian et al., 2015 (ROSE) [18], N = 238 (Placebo n = 79 vs. Rontalizumab n = 159). Subjects with moderate to severe SLE as defined by BILAG ^b^ index. 75.6% of patients had baseline high interferon signature.	II	There was no significant treatment difference in BILAG ^c^ Index Response and SRI(4) ^c^ in rontalizumab and placebo groups.	Baseline IFN signature was stratified by gene expression of a 3-gene set of IFN-regulated genes. The patients with baseline low IFN signature appeared to be the most responsive to rontalizumab and had, on average, lower anti-dsDNA titres and less profound hypocomplementemia (C3, C4) compared with the patients with baseline high IFN signature.	The incidence of serious adverse effects was comparable between the placebo and rontalizumab.
Merrill et al., 2011 [70], N = 51 (Placebo n = 17 vs. Sifalimumab n = 34). Patients with mild to moderate SLE as defined by SLEDAI score and BILAG index. ~58% of patients had baseline high IFN signature.	I	Consistent trends of greater improvement in the sifalimumab group were found using different measures, although statistical significance was not reported. Sifalimumab subjects were less likely to exhibit a SLEDAI flare or a BILAG flare.	Baseline type I IFN high or low signature statuses were determined by expression levels of 21 type I IFN-inducible genes using RT-PCR. Sifalimumab was found to neutralize overexpression of type I IFN signature in a dose-dependent manner.	Adverse event rates were similar among groups and were mostly mild. No relationship was apparent between Sifalimumab dose and severity or frequency of adverse events.
Petri et al., 2013 [71], N = 161 (Placebo n = 40 vs. sifalilumab n = 121). Most patients had moderate to severe SLE, as defined by SLEDAI score of ≥6. 75.2% of subjects had baseline high IFN signature.	I	Serum sifalimumab concentrations increased in a linear and dose-proportional manner. No statistically significant differences in clinical activity, as measured by SLEDAI and BILAG between sifalimumab and the placebo, were observed. However, when adjusted for excess burst steroids, SLEDAI change from baseline showed a positive trend over time. A trend toward normal complement C3 or C4 level at week 26 was seen in the sifalimumab groups compared with baseline.	At baseline, patients were categorized by type I IFN–inducible gene signature from a panel of 21 type I IFN–inducible genes. Dose-dependent neutralization of the type I IFN gene signature (21-gene panel) in the blood with sifalimumab treatment was observed in patients who had overexpression of the type I IFN signature at baseline. Patients with a baseline high IFN signature showed a greater mean reduction from baseline in SELENA–SLEDAI score in the combined sifalimumab group compared with the placebo group. Inhibition of the type I IFN by sifalilumab was found to be dose-dependent.	The frequencies of severe adverse effects were similar between the two treatment groups, with no apparent dose effects across the individual sifalimumab dose groups. Most adverse effects were mild or moderate and the most frequent treatment-related adverse effects were urinary tract infection, nausea and headache. Sifalilumab was generally well-tolerated.
Khamashta et al., 2016 [72], N = 431 (Placebo n = 108 vs. sifalilumab n = 323). Patients with moderate to severe SLE as defined by SLEDAI-2K ^d^ and BILAG scores. ~81% of subjects had baseline high IFN signature.	IIb	Compared with placebo, a greater percentage of patients who received sifalimumab met the primary end point of SRI (4). Improvements were consistent across various clinical end points, including global and organ-specific measures of disease activity.	Baseline IFN signature was measured based on expression of four type I IFN-regulated genes.Substantial improvements in SRI (4) and BICLA ^e^ were observed for baseline IFN-high patients vs. placebo. A meaningful statistical comparison between patients based on baseline IFN signature low vs. high was not possible due to the small number of patients with baseline low IFN gene signature.	Adverse events occurred with similar frequencies in the sifalimumab and placebo groups, except that herpes zoster infections were more frequent with sifalimumab treatment. The incidence of adverse effects was similar between sifalimumab and placebo groups.
Tummala et al., 2018 [73], N = 30. Anifrolumab in healthy subjects.	I	Anifrolumab reached maximum serum concentration after 4–7 days and were below the limit of detection by day 84 of administration. Subcutaneous administration of anifrolumab 300 mg and 600 mg exhibited dose-proportional pharmacokinetics.	-	Anifrolumab was generally safe and well-tolerated.
Furie et al., 2017 (MUSE) [74], N = 305 (Placebo n = 102 vs. anifrolumab n = 203). Patients with moderate to severe SLE, as determined by SLEDAI-2K and BILAG scores. ~75% of subjects had baseline high IFN signature.	IIb	A greater proportion of subjects treated with anifrolumab exhibited an SRI(4) response at week 24 than subjects who received placebo. Anifrolumab-treated patients had greater improvements in organ-specific and global outcomes compared to the placebo.	Baseline IFN signature was measured based on expression of four type I IFN-regulated genes. Greater efficacy of anifrolumab was found in subjects with a baseline high IFN signature as compared to the placebo. In the baseline high IFN signature subpopulation of patients, anifrolumab was found to be effective in supressing type I IFN gene expression. The median neutralization ratio was 89.7 and 91.7 for anifrolumab 300 mg and 600 mg, respectively. No neutralization was observed with the placebo.The efficacy of anifrolumab in subjects with baseline low IFN signature was similar to that of the placebo group.	Anifrolumab was well-tolerated, and incidence of adverse events was similar in anifrolumab and placebo groups. However, a dose-related increase in respiratory tract infections such as herpes zoster and influenza was observed.
Furie et al., 2019 (TULIP-1) [17], N = 457 (Placebo n = 184 vs. anifrolumab n = 273). Patients with moderate to severe SLE as determined by SLEDAI-2K score. ~82% of subjects had baseline high IFN signature.	III	The proportion of patients with an SRI(4) response was initially similar between anifrolumab and placebo. Following adjustment of medication rules, key analyses were reperformed and anifrolumab was found to improve organ-specific measures, BICLA response and sustained oral corticosteroid dose reduction.	Baseline IFN signature was measured based on expression of four type I IFN-regulated genes. In patients with high baseline IFN signature, a 21-gene assay assessing neutralization of type I IFN-induced transcripts found that anifrolumab 300mg caused early suppression (median percentage of baseline signature at week 12 was 12·6%), which was sustained through 52 weeks. No IFN gene signature suppression was observed with the placebo.	Anifrolumab was well-tolerated and had an acceptable safety profile. Incidence of adverse events was similar across groups.
Morand et al., 2019 (TULIP-2) [8], N = 362 (Placebo n = 182 vs. anifrolumab n = 180). Patients with moderate to severe, active SLE as determined by SLEDAI-2K score. ~83% of subjects had baseline high IFN signature.	III	Monthly administration of anifrolumab was found to result in a significantly higher proportion of patients with a BICLA response than the placebo. Anifrolumab treatment was also associated with reductions in oral glucocorticoid dose, severity of skin disease and counts of swollen and tender joints.	Baseline IFN signature was measured based on expression of four type I IFN-regulated genes. A 21-gene assay assessing neutralization of type I IFN-induced transcripts found that anifrolumab achieved neutralization of IFN signature early in treatment of patients with baseline high IFN signature. Among patients with baseline high IFN signature group, the percentage of patients treated with anifrolumab with a BICLA response at week 52 was significantly higher than those in the placebo group.	Anifrolumab was generally safe and well-tolerated. Incidence of herpes zoster was found to be increased in the anifrolumab group, as compared to the placebo group.
Jayne et al.,2022 (TULIP-LN) [75], N = 145, (Placebo n = 49 vs. anifrolumab n = 98). Patients with a biopsy-proven diagnosis of class III/IV ± V LN. ~94.8% of subjects had baseline high IFN signature.	II	The primary end point of a change in baseline 24-h UPCR for the combined anifrolumab compared to the placebo group at week 52 was not met. The intensified regimen of anifrolumab was associated with numerical improvements across various secodnary end points, such as percentage of subjects attaining aCRR and sustained glucocorticoid reduction.	Baseline type I IFN high or low signature statuses were determined by expression levels of 21 type I IFN-inducible genes using RT-PCR. In patients with high baseline IFN signature, the 21-gene assay found that anifrolumab displayed a dose-dependent neutralization of >80% of IFN signature. No IFN gene signature suppression was observed with the placebo.	Anifrolumab was safe and well-tolerated. The safety profile in LN patient was found to be similar to that in SLE patients without active renal disease.

^a^ Safety of Estrogens in Lupus Erythematosus National Assessment version of the SLE Disease Activity Index. ^b^ British Isles Lupus Assessment Group. ^c^ Systemic Lupus Erythematosus Responder Index (4). ^d^ Systemic Lupus Erythematosus Disease Activity Index-2000. ^e^ British Isles Lupus Assessment Group (BILAG)-Based Composite Lupus Assessment.

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
