# Peer review of "Type I Interferons in Systemic Lupus Erythematosus: A Journey from Bench to Bedside"

_ijms, 2022, doi:10.3390/ijms23052505_

Round 1

Reviewer 1 Report

In this narrative review, the Authors nicely discuss the biologic background regarding the IFN pathways leading to SLE and analyze the results of RCT published up to now with a prticular emphasis about anifrolumab.

The paper is well written, and the most relevant issues related to the therapeutic approach focused on inhibition of IFN pathways in SLE are covered and well discussed.

Although the Authors already mention (see paragraph n. 2) the relationships between anti-RBP antibodies and IFN gene signature, I suggest to add a new reference related to a recently published paper [1] focusing on this topic.

[1] Anti-RNP antibodies are associated with the interferon gene signature but not decreased complement levels in SLE. Erika L Hubbard, David S Pisetsky, Peter E Lipsky (2022-02-03). 10.1136/annrheumdis-2021-221662

Reviewer 2 Report

In this manuscript entitled “Type I Interferons in Systemic Lupus Erythematosus: A Journey From Bench to Bedside”, the authors describe the discovery of interferons, their potential contribution of type 1 interferon in SLE (but not limited to type 1 only), and association with clinical features. Finally, the results of clinical trials targeting type 1 IFN in SLE are also well summarized. The manuscript is well-written and deserves publication in its present format.

I would only like to recommend the authors to provide the results of anifrolumab treatment in patients with lupus nephritis, which was published recently (Phase II randomised trial of type I interferon inhibitor anifrolumab in patients with active lupus nephritis (PMID: 35144924)).
